# First Molecular Identification of *Caligus clemensi* on Cultured Crimson Snapper *Lutjanus erythropterus* on Jerejak Island, Penang, Peninsular Malaysia

**DOI:** 10.3390/pathogens11020188

**Published:** 2022-01-29

**Authors:** Zary Shariman Yahaya, Mohd Nor Siti Azizah, Luay Alkazmi, Rajiv Ravi, Oluwaseun Bunmi Awosolu

**Affiliations:** 1School of Biological Sciences, Universiti Sains Malaysia, Penang 11800, Malaysia; rajiv_ravi86@yahoo.com (R.R.); awosolu@student.usm.my (O.B.A.); 2Institute of Marine Biotechnology, Universiti Malaysia Terengganu, Kuala Terengganu 21030, Malaysia; sazizah@usm.my; 3Department of Biology, Umm Al-Qura University, Makkah 21961, Saudi Arabia; lmalkazmi@uqu.edu.sa; 4Department of Biology, Federal University of Technology, Akure 704, Nigeria

**Keywords:** aquaculture industry, aquaculture, fish parasites, morphological identification, phylogenetic tree

## Abstract

Fish parasites such as *Caligus clemensi* are a serious concern for cultured fish in many regions of the world, including Malaysia. This study was designed to elucidate the parasites’ prevalence and intensity coupled with the morphology and molecular identification of *C. clemensi* on cultured *Lutjanus erythropterus* in Jerejak Island, Penang, Peninsular Malaysia. The study was carried out on 200 fish specimens of cultured *L. erythropterus* obtained from the GST group aquaculture farm. Parasites were collected from the infested part of *L. erythropterus* fish, and their prevalence and intensity were determined. The parasites were identified morphologically using a field emission scanning electron microscope. Molecular studies were performed through PCR amplification and sequencing. MEGA 5 was used to construct a phylogenetic tree using the pairwise distance method. The results showed that only the *C. clemensi* parasite was found prevalent on *L. erythropterus* fish with a prevalence and mean intensity (S.D) of 198 (99%) and 36.4 ± 12.2, respectively. The prevalence varied significantly with respect to fish length (*p* < 0.05). The nucleotide BLAST sequence for 18S ribosomal RNA partial sequences showed 97% with 100% query similarity, E-value 0 with *C. clemensi* with the accession number DQ123833.1. Conclusively, *C. clemensi* remains a major parasite of *L. erythropterus* in the study area.

## 1. Introduction

The aquaculture system in floating cages was first established in Penang, Malaysia in 1973. Meanwhile, the snapper fish (Lutjanidae) was the commonest species among marine fish cultures reared in the aquaculture industry [1]. Presently, it is still cultured in many Asian countries including Malaysia. The development of marine cultures has been linked to the emergence of parasitic diseases occasioned by ectoparasites which generally affect fish production [2]. Due to its high infestation rate, the *Caligus* species of copepods are an important threat to the marine culture industry [3]. Thus, diseases caused by parasitic copepods such as Caligids have become widely distributed throughout cultured marine snapper (*Lutjanus* spp.) and other fish species. The *Caligus* species is responsible for causing massive mortalities in cultured *L. erythropterus* due to pathological tissue alterations on the gills and buccal cavity, thereby leading to serious economic loss. They cause serious harm on the host fishes when they feed on the epidermal tissue, blood and mucus of fishes [1,3]. Previous studies in Malaysia have shown that there were five species of copepod from *Caligus* general found in marine farmed fishes in Penang and Langkawi Island, according to Maran et al. [3]. These include *Caligus chiastos* from golden snapper, *Lutjanus johni* (Bloch); *Caligus epidemicus* from sea bass, *Lates calcarifer* and *Epinephelus coioides*; *Caligus longipedis* from *Gnathonodon speciosus*; *Corydoras punctatus* from *L. calcarifer*; and *Caligus rotundigenitalis* from *L. erythropterus*, *Epinephelus bleekeri*, *E. fuscoguttatus* and *G. speciosus*. Similarly, a study conducted by Leaw et al. [1] reported a high parasitic infestation rate of 81% in *L. erythropetrus* fish from floating cages off of Penang Island. Besides, reports have indicated that this parasite has a significantly high potential to affect the fish’s growth, productiveness and survival rate [4].

Though some findings on the prevalence, mean intensity and veterinary problems caused by the *Caligus* species of copepods have been reported [1,3], the molecular identification of the species is still lacking [1]. Thus, this study is novel since it is the first study to carry out the molecular identification of *C. clemensi* in Malaysia. Moreover, there is a need for adequate updated epidemiological information on the prevalence of the *Caligus* species in Penang to evaluate the impact of *C. clemensi* on cultured crimson snapper and take appropriate action for its control. With this aim, the morphological and molecular identifications of *C. clemensi* on cultured crimson snapper were performed to elucidate the parasite prevalence and mean intensity. In this study, *C. clemensi* was identified for the first time in Malaysia with species confirmation by molecular techniques.

## 2. Material and Methods

### 2.1. Parasite Examination on Crimson Snapper

All experimental procedures involving animals were conducted at the Aquatic Research Complex L24, USM, Penang from February to May 2016, in accordance with ethical and practical guidelines [5]. The research was approved by the Universiti Sains Malaysia Animal Ethics Committee (reference number: 1001/PBIOLOGI/855003). The survey was carried out on 200 fish specimens of cultured *L. erythropterus* fish species from aquaculture farm GST group Sdn Bhd on Jerejak Island, Penang, Peninsular Malaysia (5°18′53.2′′ N 100°19′29.2′′ E). The fish length (cm) was measured prior to parasite examination and categorized into 26–28 cm, 29–31 cm and 32–35 cm according to the authors in [1]. Generally, the fish length was categorized based on growth and development of the fishes with respect to size such that it included the smallest (26–28 cm), the medium (29–31 cm) and the largest (32–35 cm). A solution was prepared containing fresh water and tricaine methane-sulfonate (MS-222) (Sigma–Aldrich) (50 mg/L) to anaesthetize the fish and reduce handling stress. The freshwater medium was used as an anesthetic to reduce the stress as well as for easy handling [6,7]. After the fish had been anaesthetized, parasites were collected from the external skin surfaces of infested areas of the fish such as the head, body and both sides of the inner operculum using surgical dissecting forceps [6,7]. The fish parasites were obtained and measured. All of the *C. clemensi* parasites obtained from the 200 fish samples were calculated based on a morphological field examination [8,9] with the aid of a dissecting microscope (magnification of X50; Leica, Allendale, NJ, USA). The prevalence and mean intensity were calculated according to the method of Bush et al. [10]. The parasite prevalence was defined as the total number (percentage) of infected fishes while mean intensity was defined as the average number of parasites in an infected fish.

### 2.2. Morphological Identification Using Scanning Electron Microscope

Secondary morphological identification was performed using a Supra 50 VP ultra- high-resolution LEO analytical field emission scanning electron microscope (Carl Zeiss LEO Supra 50 VP field emission equipped with an Oxford INCA system). Sample preparation was carried out according to the Carl Zeiss LEO Supra 50 VP field emission scanning electron microscope manufacturer’s procedures and protocol. First, parasite samples were immersed in ethanol and hydrated with 90%, 80% and 70% serial dilutions. After that, the parasite specimens were placed on carbon film-coated 400 mesh copper grids for 1–3 min. Then, filter paper was used to dry the specimens. Grids were then placed in a desiccator using filter paper-lined petri plates. After three days of preservation, imaging was completed [11,12]. The identification of parasites collected was made by a morphological inspection using appropriate identification keys after the parasite pictures were obtained [3,8,9,11,13,14,15].

### 2.3. Molecular Identification

The DNA extraction and purification of each parasite sample were performed using the Qiagen DNeasy blood and tissue kit (Qiagen, Inc., Valencia, CA, USA) following the manufacturer’s instructions. Purified genomic DNA was eluted by adding 100 uL of elution buffer AE to the same spin column in a new eppendorf tube and centrifuged at 5200 g for 1 min. The centrifuge step was repeated for a total of 200 uL of the sample volume. DNA sample concentration and quality were measured using an ACT-Gene NanoDrop spectrophotometer (ASP 2680, Taipei, Taiwan). Eluted genomic DNA was stored at −20 °C until the PCR analysis. Subsequently, 18S ribosomal partial sequences were amplified from purified genomic DNA using the Universal Folmer primers LCO 1490 (5′-GGT CAA CAA ATC ATA AAG ATA TTG G-3′) as a forward primer and HCO2198 (5′-TAA ACT TCA GGG TGA CCA AAA AAT CA-3′) as a reverse primer [16,17]. A polymerase chain reaction (PCR) was carried out using a total volume of 25 µL master mix solutions (14 µL of ddH_2_O, 2.5 µL of Promega PCR buffer, 3 µL of Promega MgCl_2_ solutions, 1 µL of Promega dNTP, 1 µL of each forward and reverse primers, 2 µL of DNA template and 0.5 µL of Promega Go Taq DNA polymerase). Standard cycle conditions for the PCR were set accordingly by initial denaturation of 10 min at 95 °C, followed by 35 cycles of 30 s at 95 °C, 30 s at 50 °C, 60 s at 72 °C and final elongation of 7 min at 72 °C. The PCR was carried out in Mycycler thermal cycler Bio-Rad PCR systems (Hercules, CA, USAUSA). Purification of the PCR product was performed using the procedure and materials provided in a QIAquick PCR purification kit (Qiagen, Inc.). Amplification products were sequenced in both directions.

A distance-based tree approach to species identification was conducted using MEGA 5 software (version 5). A BLAST search was conducted with a DNA sequence that was amplified. Using MEGA 5, a distance-based tree approach to species identification was carried out by neighbor-joining the 18S ribosomal partial sequences of the recorded species from the BLAST search and those analyzed in this study. A pairwise distance calculation was completed using MEGA 5 analysis tools and the method of Tajima and Nei [18] served as the substitution model [19]. In addition, the bootstrap method was deployed as a test of phylogeny using 1000 bootstrap replications. DNA barcoding was developed according to bold system web instructions. A character-based diagnostic approach to species delimitation was applied using MEGA 5.

### 2.4. Statistical Analysis

The statistical analysis was performed using the Statistical Package for Social Sciences software, SPSS version 20. A chi-square analysis was used to compare parasite prevalence. Values at *p* < 0.05 were considered significant.

## 3. Results

### 3.1. Prevalence and Mean Intensity of Caligus clemensi

A total of 200 crimson snapper fishes were examined for the prevalence of *C. clemensi*. The fishes were categorized into 26–28 (25%), 29–31 (49%) and 32–35 (26%). The results showed a total prevalence and mean intensity (S.D) of 99% (198) and 36.4 ± 12.2, respectively. Prevalence according to fish length showed that fishes with lengths 26–28 cm and 32–35 cm had a parasite prevalence of 100%, while fishes with 29–31 cm had 97.96%. The result was statistically significant (*p* < 0.05). Similarly, the mean intensity significantly (*p* < 0.05) increased with an increase in fish length such that fishes with 26–28 cm had the lowest mean intensity of 29.48 ± 11.33, followed by fishes with 29–31 cm with a mean intensity of 33.32 ± 10.81 and fishes with the longest length of 32–35 cm had the highest mean intensity of 48.77 ± 4.0 (Table 1).

### 3.2. Morphological Analysis of Caligus clemensi

A total of twenty [20] *C. clemensi* were isolated for examination under the scanning electron microscope (SEM). Each of the *C.*
*clemensi* parasites was isolated and viewed on an optical microscope (Figure 1) for the initial minor observations while the detailed observations were observed with a scanning electron microscope (SEM) (Figure 2; Appendix A). Overall, the total body length was 3.9 to 4.1 mm (Figure 2A). The cephalothorax shield was ovate, with a length of 2.5 to 2.7 mm greater than the width of 1.9 to 2.2 mm (Figure 2A). Furthermore, the mouth tube was carried folded parallel to the body axis (Figure 2D), which showed the tip of the labium for a typical *C. clemensi*. The fourth leg bearing segment measured from 0.3 to 0.2 mm and was located posterior to the cephalothorax (Figure 2B). The genital complex was sub-circular to oval, measuring 0.9 to 1.0 mm (Figure 2A). The abdomen was divided into two segments that were longer than they were wide, measuring 0.7 to 1.0 mm in length (Figure 2A). Leg 4 was positioned on both sides of the genital complex curve (Figure 2E). The furca was carried in a folded position parallel to the body axis, displaying its shape as a flat surface (Figure 2F). All the other organs were labelled as shown in Figure 3 and Figure 4.

### 3.3. Molecular Analysis of Caligus clemensi

All the sequences were successfully analyzed and recovered from all *C. clemensi* individuals without any stop codons. The nucleotide BLAST sequence for 18S ribosomal RNA partial sequences showed 97% with 100% query similarity, E-value 0 with *C. clemensi* DQ123833.1 (Figure 5). Moreover, all the sequences from this study have been deposited in GenBank with accession numbers KX808655, KX808656 and KX808657. Additionally, the barcode of Life Data System were obtained and the BOLD submission ID in accordance with species *Caligus clemensi* as shown in Table 2.

Figure 6 shows the constructed phylogenetic tree between species. All of the *C. clemensi* identified in this study shared a close clade relationship with the *C. clemensi* DQ123833.1 sequence. *C. clemensi* was found to be in a closer clade than *C. centrodonti* EF088407, *C. curtus* EF088407 and *C. pelamydis* EF088411.1. This study found distant clades from *C. clemensi* in *C. uniartus* KC569363.1, *C. punctatus* KR048777.1 and *C. fugu* KR048778.1.

## 4. Discussion

This study reports the first molecular identification of *C. clemensi* on cultured crimson snapper at Jerejak Island, Penang, Peninsular Malaysia. Previous studies on *Caligus* species from marine finfish cage culture species are few and limited in Malaysia. One of the previous studies conducted by Ahamad Hasmi [20] reported the isolation and identification of three different species of caligids from *Lates calcarifer* cultured in floating net cages in Malaysia. These species were: *C. chiastos*, *C. epidemicus* and *C. rotundigenitalis* [20]. Our findings are substantiated by other studies that were previously reported [1,2,3]. The *Caligus* species, for example, is known to have a podoplean-type body structure in terms of morphology [21,22]. The *Caligus* species also has one rather than five segments between the cephalothorax shield and the genital segment [9,23]. Moreover, the *Caligus* species is known to be siphonostomatoid due to the presence of a distinctive tubular mouth apparatus. This type of mouth part is characterized by the elongated and tapering structure [21].

Furthermore, the sequence analysis and construction of phylogenetic trees presented in this study reveal the relationship between our study isolates and the others available in the GenBank (Figure 6). The molecular analyses derived from 18S ribosomal RNA partial sequences accurately supported the morphological classification of *C. clemensi*. This is corroborated by the findings of Jones et al. [24]. The phylogenetic tree of *C. clemensi* showed a closer clade of *C. centrodonti*, *C. curtus* and *C. pelamydis*, compared to the distant clade of *C. uniartus*, *C. punctatus* and *C. fugu,* and this is similar to the findings of Øines and Schram [25].

Accurate phylogenetic analyses require the stability of taxonomic classifications, and it is often concluded, based on RNA sequences, that and an increased number of data would result in a higher tree resolution [26]. The taxonomic and systematics of *Caligus* spp. are still very far from complete in their largest genus of Copepoda comprising more than 250 species: Müller, 1785. However, only *C. centrodonti*, *C. curtus*, *C. pelamydis*, *C. uniartus*, *C. punctatus* and *C. fugu* are appropriate to be included in this study. Noticeably, several other taxa, such as *C. elongates* and *C. warlandi*, available in the GenBank, have to be left out since our study was only exploring the published and verified sequences with Refseq.

Vast reports have highlighted the instability and high degree of variations in the phylogeny of *Caligus* spp. [27,28]. The uncertainty of *Caligus* spp. phylogenies is still debatable due to the discrepancy between several varying reports, regardless of whether the 18S, 28S or the mitochondrial genes have been utilized [25]. This is evident from the *C. clemensi* systematics, which revealed uncertainty and instability in the maximum likelihood and maximum parsimony distance analyses, where *C. clemensi* was found to be closely related to *C. pelamydis* as supported by our findings.

To date, only one study on *Caligus rotundigenitalis* infestation on *L. erythropterus* fish has been documented in Malaysia [1]. The report found a high prevalence of *C. rotundigenitalis L. erythropterus* fish at Bukit Tambun, Penang, Peninsular Malaysia, with an 81% prevalence. This high prevalence has been linked to the species’ continuous reproduction and high proliferation capacity throughout the year and its natural life cycle. Additionally, other studies have reported varying degrees of *C. clemensi* prevalence from other parts of the world such as Egypt [29,30,31] and Canada [32,33]. Similarly, our findings showed a total prevalence of 198 (99.00%) for *C. clemensi* on *L. erythropterus*. This is higher than the report by Leaw et al. [1] who recorded a prevalence of 81.4% for *C. rotundigenitalis*. This variation could be due to the duration of infestation on the fish, fish immune response or the sizes of the fishes examined. Other factors include host response to infestation, interaction of sea temperature, host abundance and distribution in sea cages, according to the report of Costello [34]. Moreover, it was observed in this study that fish length varied significantly with respect to parasite prevalence and mean intensity (*p* < 0.05). The longer the fish, the higher the parasite infestation. The prevalence and mean intensity of *Caligus* spp. have commonly been found to show a positive correlation to the length size of the fish host [11,35]. According to Rohde [35], more investigation revealed that different age stages of fish might have variable infestation levels due to parasite accumulation, the host immune system, size range and feeding habits. Furthermore, as the host’s immune system grows over time, the parasite’s intensity reduces.

## 5. Conclusions

Conclusively, this study has successfully identified *C. clemensi* morphologically and through the use of molecular techniques. Additionally, this study showed that crimson snapper cultured at the GST group aquaculture farm had a high prevalence and heavy infestation of *C. clemensi* with the prevalence and mean intensity of 99% and 36.4 ± 12.2, respectively. This updated information on the prevalence and mean intensity can inform appropriate control measures for the proper management of cultured crimson snapper.

## Figures and Tables

**Figure 1 pathogens-11-00188-f001:**
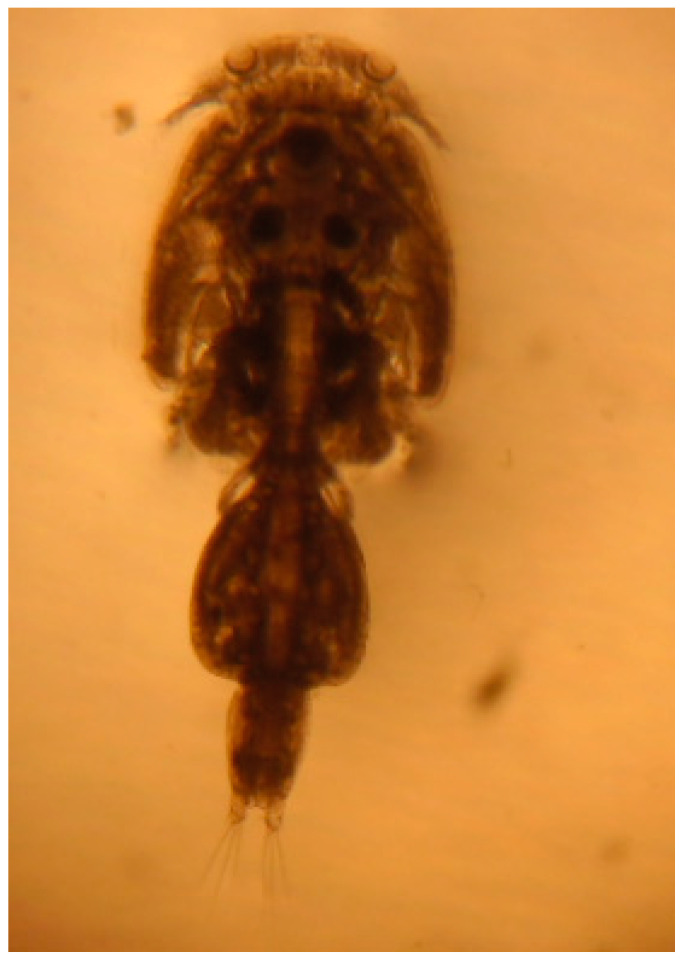
*Caligus clemensi* ventral view with the aid of optical microscope, magnification 50X.

**Figure 2 pathogens-11-00188-f002:**
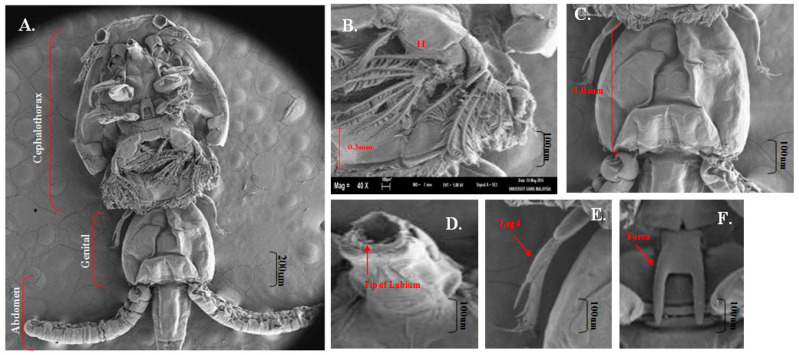
*Caligus clemensi* viewed under SEM. (**A**): Ventral view of adult female with egg sacs; (**B**): leg 2 measured as 0.3 mm; (**C**): genital complex measured as 1.0 mm; (**D**): tip of labium detailed; (**E**): leg 4 detailed; (**F**): furca detailed. Magnifications (**A**) = 35X, (**B**) = 40X, (**C**) = 40X, (**D**) = 80X, (**E**) = 50X and (**F**) = 80X.

**Figure 3 pathogens-11-00188-f003:**
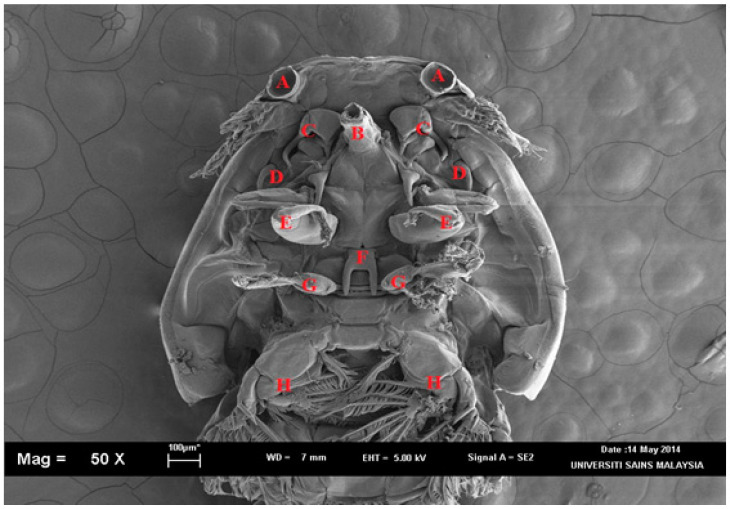
*Caligus clemensi*. in ventral view. A: lunule; B: mouth tube; C: 3-segmented antenna which consists of basal segment (=coxa), middle segment (=basis) and terminal segment (=endpod); D: 1-segmented hook-like post antennal process; E: maxillae; F: sternal furca that has tapering tines; G: leg 1; H: leg 2. Magnification at 50X.

**Figure 4 pathogens-11-00188-f004:**
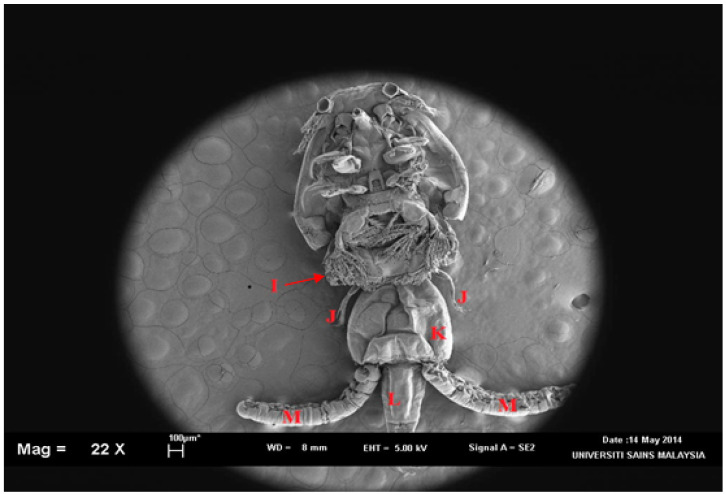
*Caligus clemensi* in ventral view. I: Leg 3; J: leg 4 with 5 setae; K: genital complexes which attach to the posterior portion of the fourth leg-bearing segment without the posterolateral processes; L: 1st abdominal somite that leads to anal somite where caudal ramus originates from it; M: egg sac. Magnification at 22X.

**Figure 5 pathogens-11-00188-f005:**
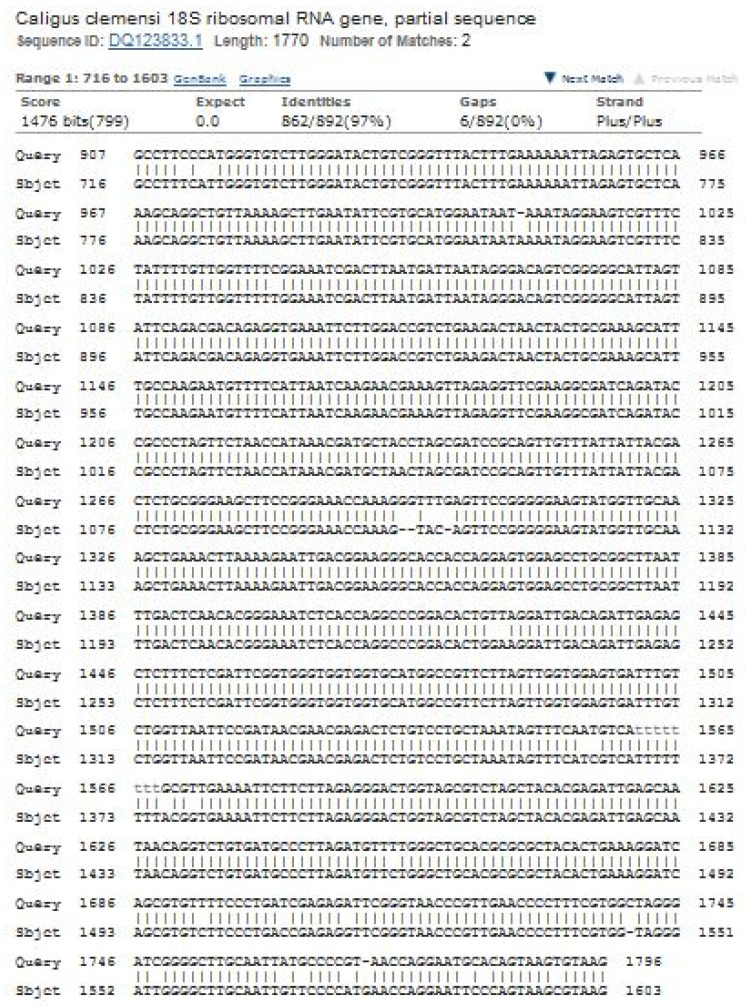
Verification of the obtained sequence of 18S ribosomal RNA partial sequences’ comparison with *Caligus clemensi* (DQ123833.1) using ClustalW, multiple sequence alignment software. There is an indication of 97% similarity between both sequences.

**Figure 6 pathogens-11-00188-f006:**
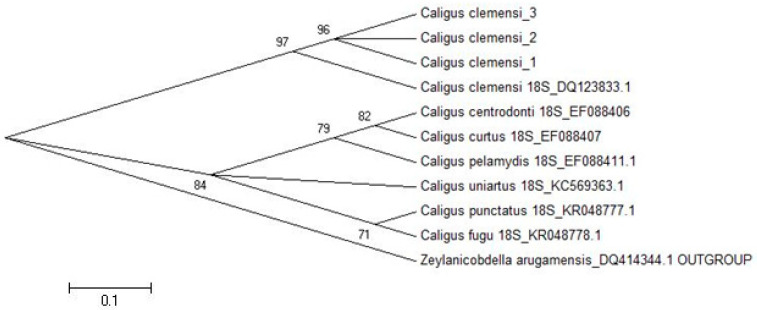
MEGA 5 generated neighbor-joining tree based upon the 18S ribosomal RNA partial sequences of specimens analyzed in this study.

**Table 1 pathogens-11-00188-t001:** Prevalence and mean intensity of *Caligus clemensi* with regards to the length of crimson snapper.

Parasite	Fish Length (cm)	Number of Fish Examined (%)	Number of Fish Infected (%)	Minimum No. of Parasites Recovered	Maximum No. of Parasite Recovered	Total Number of Parasites Recovered	Mean ± SD
* **Caligus clemensi** *	26–28	50 (25)	50 (100.00)	15	53	1474	29.48 ± 11.33
	29–31	98 (49)	96 (97.96)	15	53	3265	33.32 ± 10.81
	32–35	52 (26)	52 (100.00)	42	54	2536	48.77 ± 4.0
**Total**		200	198 (99.00)			7275	36.4 ± 12.2

**Table 2 pathogens-11-00188-t002:** Barcode of Life Data System, (www.boldsystem.org, accessed on 12 August 2017) BOLD submission ID in accordance with species *Caligus clemensi*.

Identification	Specimen Page	Sequence Page	COI-SP	BIN
*Caligus clemensi*	Fish Caligus 5	CALIG005-14	671 [0n]	BOLDAC02743
*Caligus clemensi*	Fish Caligus 4	CALIG004-14	671 [0n]	BOLDAC02743
*Caligus clemensi*	Fish Caligus 3	CALIG003-14	671 [0n]	BOLDAC02743
*Caligus clemensi*	Fish Caligus 2	CALIG002-14	671 [0n]	BOLDAC02743
*Caligus clemensi*	Fish Caligus 1	CALIG001-14	671 [0n]	BOLDAC02743

## Data Availability

The data set presented in this study are available upon reasonable request from the corresponding author.

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
