# Peer review of "First Molecular Identification of Caligus clemensi on Cultured Crimson Snapper Lutjanus erythropterus on Jerejak Island, Penang, Peninsular Malaysia"

_pathogens, 2022, doi:10.3390/pathogens11020188_

Round 1

Reviewer 1 Report

General comments

This study uses molecular markers to identify a copepod species present in crimson snappers from an aquaculture farm in Penang, Malaysia. Although an interesting topic, the study is overall poorly presented. Points of concern include unclear classification of fish sizes, collection of data that was not included in the analyses/results/discussion, vagueness regarding how the data was analyzed, and the and morphological description of a parasite species based on one individual only.

Below, I list a few other things to consider:

Keywords

  • Avoid repeating words already included in your title.
  • Please use italics when referring to scientific names.

Introduction

  • The introduction does not provide a strong background for the study aim.
  • Use full scientific name when introducing a species for the first time (Genus species) and then use abbreviated form throughout the text ( species).
  • 42-44: do you mean that the prevalence was constant throughout the 10 months or was that the study period?
  • Ln 49-50: this suggests that this is one of your goals, however you don’t mention anything related to parasite spread later in your manuscript.
  • 51-53: what is the purpose of the molecular identification? You seem to know that you’re dealing with C. clemensi already.

Methods

  • When was the study conducted?
  • Ln 61-62: how were length categories defined? Are there any physiological or ontogenetic reasons behind this?
  • Ln 64-66: where?
  • Ln 67-68: is it common practice to place sea water fish in fresh water? What is the purpose behind this? Methodology is not sufficiently developed.
  • Ln 69-71: what is the purpose of taking on site temperature and salinity?
  • 71-75: prevalence is a well understood measure, I don’t think you need to include a formula here.
  • Ln 90: what do you mean by “every parasite was separated accordingly”?

Results

  • Ln 129-130: prevalence is expressed as a percentage. You can indicate the number of infected fish in parenthesis.
  • How did you define parasite density? This was not explained in the methods.
  • Table 2 does not seem to have a purpose, as you didn’t conduct any tests including salinity and temperature.
  • Ln 147-149: am I understanding correctly that you only observed and measured one out of the 7275 collected parasites? This is definitely insufficient and does not reflect intra-species variation.

Discussion

  • Ln 203-204: this statement seems redundant, as these are in fact different species.

Conclusion

  • What is the relevance of your conclusions? From your manuscript alone, it’s not clear to me that the presence of clemensi is harmful to the crimson snapper. How does your study help in the control and management of copepods?

Author Response

Response to Reviewer 1 Comments

Thank you very much for your efforts in making this paper a wonderful manuscript through appropriate and sincere review. Our responses are highlighted below.

Keywords

Reviewer comment: Avoid repeating words already included in your title.

Authors’ response: some keywords occurring in the title have been replaced, for instance, Aquaculture, fish parasites and Penang, have been included.

Reviewer comment: Please use italics when referring to scientific names.

Authors’ response: all the scientific names have been italicized as suggested, thank you.          

Introduction

Reviewer comment: The introduction does not provide a strong background for the study aim.

Authors’ response: additional information has been provided to buttress the study aim and the additional information is:Caligus species is responsible for causing massive mortalities in cultured fishes L. erythropterus due to pathological tissue alterations on the gills and buccal cavity thereby leading to serious economic loss. They cause serious harm on the host fishes when they feed on the epidermal tissue, blood and mucus of fishes”. Further justification includes “lack of molecular identification of Caligus clemensi in Malaysia and need for updated information on the spread of the disease. All these have been included to form a strong background for the aim.

Reviewer comment: Use full scientific name when introducing a species for the first time (Genus species) and then use abbreviated form throughout the text (species).

Authors’ response: full scientific names have now been included when introduced for the first time and abbreviated subsequently as suggested. Thank you.

Reviewer comment: 42-44: do you mean that the prevalence was constant throughout the 10 months or was that the study period?

Authors’ response: Thank you for pointing this out. The period of 10 months indicated have been removed. The total prevalence of 81% was an average prevalence. It now read as “Leaw et al. (1) reported a high parasitic infestation rate of 81% on L. erythropetrus fish from floating cages of Penang Island”

Reviewer comment: Ln 49-50: this suggests that this is one of your goals, however you don’t mention anything related to parasite spread later in your manuscript.

Authors’ response: yes, one of our goals is to determine the parasite prevalence. Meanwhile the word “spread” has been replaced by “prevalence”. The prevalence was well highlighted in line 265-282.

Reviewer comment: 51-53: what is the purpose of the molecular identification? You seem to know that you’re dealing with C. clemensi already.

Authors’ response: the purpose of molecular identification in this study is to properly detect the parasites, differentiate and compared to others in the same genus or family. Moreso that molecular diagnosis is very sensitive compared to other methods of diagnosis including microscopy. Also, phylogenetic tree classifying the parasite with respect to others was done.

Methods

Reviewer comment: When was the study conducted?

Authors’ response: The study was conducted between February to May, 2016 and this is now included in the methodology.

Reviewer comment: Ln 61-62: how were length categories defined? Are there any physiological or ontogenetic reasons behind this?

Authors’ response: Yes, the fish length was categorised based on the ontogenetic reasons. The minimum, average and maximum length were classified accordingly, based on the general size of the fish caught. Thank you.

Reviewer comment: Ln 64-66: where?

Authors’ response: the study was conducted at the Aquatic Research Complex L24, USM, Penang and this has been included in the methodology. Thank you.

Reviewer comment: Ln 67-68: is it common practice to place sea water fish in fresh water? What is the purpose behind this? Methodology is not sufficiently developed.

Authors’ response: the purpose behind placing sea water fish in fresh water is to anaesthetise the fish since fresh water medium can be used as anaesthetics to reduce the stress as well as for easy handling. This statement has been included in the methodology. Thank you.

Reviewer comment: Ln 69-71: what is the purpose of taking on site temperature and salinity?

Authors’ response: we initially wanted to see the effect of temperature and salinity of the fish but did not have any significant effect. Thus, the statement about the salinity and temperature have been removed. The table has also been removed.

Reviewer comment: 71-75: prevalence is a well understood measure, I don’t think you need to include a formula here.

Authors’ response: formula have been removed. Thank you for your suggestion.

Reviewer comment: Ln 90: what do you mean by “every parasite was separated accordingly”?

Authors’ response: the statement has been removed since it does not have any significant meaning. Thank you

Results

Reviewer comment: Ln 129-130: prevalence is expressed as a percentage. You can indicate the number of infected fish in parenthesis.

Authors’ response: the number of infected fish has been indicated in parenthesis

Reviewer comment: How did you define parasite density? This was not explained in the methods.

Authors’ response: the parasite density has been replaced with a better word which is mean intensity. Mean intensity is the average number of parasites in an infected fish.

Reviewer comment: Table 2 does not seem to have a purpose, as you didn’t conduct any tests including salinity and temperature.

Authors’ response: table 2 has been removed. thank you.

Reviewer comment: Ln 147-149: am I understanding correctly that you only observed and measured one out of the 7275 collected parasites? This is definitely insufficient and does not reflect intra-species variation.

Authors’ response: No, each fish was observed and the record was taken. Not one fish was observed. That was an error. The statement has been corrected to reflect Each of the C. clemensi parasites was isolated and viewed on an optical microscope (Figure 1) for the initial minor observations while the detailed observations were observed with Scanning Electron Microscope (SEM) (Figure 2) Thank you.

Discussion

Reviewer comment: Ln 203-204: this statement seems redundant, as these are in fact different species.

Authors’ response: the statement “However, these species are quite different from C. clemensi identified in this study” have now been removed since there is no need of stating such again.

Conclusion

Reviewer comment: What is the relevance of your conclusions? From your manuscript alone, it’s not clear to me that the presence of clemensi is harmful to the crimson snapper. How does your study help in the control and management of copepods?

Authors’ response: the conclusion has been rewritten to reflect the identification, prevalence and mean intensity of C. clemensi. The harmful effect of the parasite in terms of the pathology was not part of the objective of this study. However, the harmful effect could be a major justification for this study. Also, once the C. clemensi prevalence and intensity is known, appropriate step to guide and control C. clemensi can be taken.

Reviewer 2 Report

The article First molecular identification of Caligus clemensi on cultured Crimson Snapper Lutjanus erythropterus at Jerejak Island, Penang, Peninsular Malaysia describes the prevalence and identification of a common ectoparasite on a cultured fish species. The morphological and molecular identification revealed a high prevalence of only one species: Caligus clemensi.

The work has certain interest, since fish parasites are a serious problem for fishing farming, as well as for wild populations of fish. However, in my opinion, the relevance of the research is not clearly explained in the manuscript, the methodology used to attain the purpose of the study is not properly justified and the discussion lacks of structure and fundament.

There are several issues in the experimental design that would need more detail. For example: Why did the authors choose these three size class? Why the authors did not measure fish weight?  Why did you measure temperature and sanity from February to March? Did you perform the experiment in these months? Could it be important for your findings?

In the conclusion section is stated for the first time in the paper that “Therefore, the information gathered from this study can help in the effective control and management of ectoparasitic copepod infection in floating cages”. How your study can help to this management? What other evidences or works exist? This should be mentioned at the beginning, in the introduction.

Also, the manuscript needs further English proofreading and style must be improved. The scientific name of species must be corrected to be always in italics.

Therefore, I will not recommend this paper for publication in Pathogens.

Author Response

Response to Reviewer 2 Comments

Thank you very much for your efforts and the wonderful observations highlighted in our manuscript.

Our responses are stated below.

Reviewer comment: The article First molecular identification of Caligus clemensi on cultured Crimson Snapper Lutjanus erythropterus at Jerejak Island, Penang, Peninsular Malaysia describes the prevalence and identification of a common ectoparasite on a cultured fish species. The morphological and molecular identification revealed a high prevalence of only one species: Caligus clemensi.

Authors’ response: yes, the study identification revealed only one species, Caligus clemensi and it is the focus of this study.

Reviewer comment: The work has certain interest, since fish parasites are a serious problem for fishing farming, as well as for wild populations of fish. However, in my opinion, the relevance of the research is not clearly explained in the manuscript, the methodology used to attain the purpose of the study is not properly justified and the discussion lacks of structure and fundament.

Authors’ response: the relevance of this study is the molecular and morphological identification which is very important for classification. This study is very novel in Malaysia since no previous molecular study on C. clemensi has been carried out. Moreover, this study determined prevalence and intensity which provide updated epidemiological information on the status of C. clemensi in Malaysia. The updated information can help to know whether control action should be taking or not in that fish farm. Additional information has been included in the introduction to justify the relevance of the study. As for the methodology and discussion, specific additional information to justify each procedure have been included.

Reviewer comment: There are several issues in the experimental design that would need more detail. For example: Why did the authors choose these three-size class? Why the authors did not measure fish weight?  Why did you measure temperature and sanity from February to March? Did you perform the experiment in these months? Could it be important for your findings?

Authors’ response: the details needed in the experimental design have been included. For example, the three-size classification were selected based on the length of the fishes. The fish length was categorised based on the ontogenetic reasons. The minimum, average and maximum length were classified accordingly, based on the general size of the fish caught.

Concerning the fish weight, the measuring weight was later observed to be faulty and was giving a wrong data which cannot be applied. We are really so sorry about this. It forms part of our limitation. Please, do not allow this to hinder manuscript from publication. Please focus on the prevalence and mean intensity with respect to fish length. Thank you.

Concerning the temperature and salinity, the data and information about this have been removed since it is not important and no positive correlation or significant relationship exist. Thank you.

Reviewer comment: In the conclusion section is stated for the first time in the paper that “Therefore, the information gathered from this study can help in the effective control and management of ectoparasitic copepod infection in floating cages”. How your study can help to this management? What other evidences or works exist? This should be mentioned at the beginning, in the introduction.

Authors’ response: the statement has been removed and replaced. the conclusion has been rewritten to reflect the identification, prevalence and mean intensity of C. clemensi which are other important information elucidated in the study. It can inform appropriate control measure in the sense that when there is information on the status of C. clemensi parasite, it will help to know whether to apply treatment or not.

Reviewer comment: Also, the manuscript needs further English proofreading and style must be improved. The scientific name of species must be corrected to be always in italics.

Authors’ response: The manuscript has now been properly edited by an English-native editor.

Therefore, I will not recommend this paper for publication in Pathogens.

Authors’ response: since the corrections have been done through additional information included in the manuscript to justify area of concern, kindly reconsider your decision in order to publish this manuscript. Thank you very much for your consideration.

Please let me know other concerns so as to address it as appropriate. Thank you very much for your sincere support.

Round 2

Reviewer 1 Report

Response to Reviewer 1 Comments

Thank you very much for your efforts in making this paper a wonderful manuscript through appropriate and sincere review. Our responses are highlighted below.

Keywords

Reviewer comment: Avoid repeating words already included in your title.

Authors’ response: some keywords occurring in the title have been replaced, for instance, Aquaculture, fish parasites and Penang, have been included.

Reviewer’s response: “Lutjanus erythropterus” and “Penang” are still present in both title and abstract. I suggest you to avoid repetition so that you can add other keywords that might help you reach more readers through journal databases.

Reviewer comment: Please use italics when referring to scientific names.

Authors’ response: all the scientific names have been italicized as suggested, thank you.          

Introduction

Reviewer comment: The introduction does not provide a strong background for the study aim.

Authors’ response: additional information has been provided to buttress the study aim and the additional information is:Caligus species is responsible for causing massive mortalities in cultured fishes L. erythropterus due to pathological tissue alterations on the gills and buccal cavity thereby leading to serious economic loss. They cause serious harm on the host fishes when they feed on the epidermal tissue, blood and mucus of fishes”. Further justification includes “lack of molecular identification of Caligus clemensi in Malaysia and need for updated information on the spread of the disease. All these have been included to form a strong background for the aim.

Reviewer comment: Use full scientific name when introducing a species for the first time (Genus species) and then use abbreviated form throughout the text (species).

Authors’ response: full scientific names have now been included when introduced for the first time and abbreviated subsequently as suggested. Thank you.

Reviewer comment: 42-44: do you mean that the prevalence was constant throughout the 10 months or was that the study period?

Authors’ response: Thank you for pointing this out. The period of 10 months indicated have been removed. The total prevalence of 81% was an average prevalence. It now read as “Leaw et al. (1) reported a high parasitic infestation rate of 81% on L. erythropetrus fish from floating cages of Penang Island”

Reviewer comment: Ln 49-50: this suggests that this is one of your goals, however you don’t mention anything related to parasite spread later in your manuscript.

Authors’ response: yes, one of our goals is to determine the parasite prevalence. Meanwhile the word “spread” has been replaced by “prevalence”. The prevalence was well highlighted in line 265-282.

Reviewer comment: 51-53: what is the purpose of the molecular identification? You seem to know that you’re dealing with C. clemensi already.

Authors’ response: the purpose of molecular identification in this study is to properly detect the parasites, differentiate and compared to others in the same genus or family. Moreso that molecular diagnosis is very sensitive compared to other methods of diagnosis including microscopy. Also, phylogenetic tree classifying the parasite with respect to others was done.

Reviewer’s response: if so, then perhaps is better that you change “C. clemensi” for “Caligus spp.” in ln 52-53. As it is, it appears you already know the species you’re dealing with and, since you don’t explicitly mention that you want to compare with other Caligus populations, the importance of the molecular identification in this study is not clear.

Methods

Reviewer comment: When was the study conducted?

Authors’ response: The study was conducted between February to May, 2016 and this is now included in the methodology.

Reviewer comment: Ln 61-62: how were length categories defined? Are there any physiological or ontogenetic reasons behind this?

Authors’ response: Yes, the fish length was categorised based on the ontogenetic reasons. The minimum, average and maximum length were classified accordingly, based on the general size of the fish caught. Thank you.

Reviewer’s response: please clarify which ontogenetic reasons were behind this categorization in the text.

Reviewer comment: Ln 64-66: where?

Authors’ response: the study was conducted at the Aquatic Research Complex L24, USM, Penang and this has been included in the methodology. Thank you.

Reviewer’s response: my question was regarding the parasite collection. Where were the infested areas of the fish?

Reviewer comment: Ln 67-68: is it common practice to place sea water fish in fresh water? What is the purpose behind this? Methodology is not sufficiently developed.

Authors’ response: the purpose behind placing sea water fish in fresh water is to anaesthetise the fish since fresh water medium can be used as anaesthetics to reduce the stress as well as for easy handling. This statement has been included in the methodology. Thank you.

Reviewer comment: Ln 69-71: what is the purpose of taking on site temperature and salinity?

Authors’ response: we initially wanted to see the effect of temperature and salinity of the fish but did not have any significant effect. Thus, the statement about the salinity and temperature have been removed. The table has also been removed.

Reviewer comment: 71-75: prevalence is a well understood measure, I don’t think you need to include a formula here.

Authors’ response: formula have been removed. Thank you for your suggestion.

Reviewer comment: Ln 90: what do you mean by “every parasite was separated accordingly”?

Authors’ response: the statement has been removed since it does not have any significant meaning. Thank you

Results

Reviewer comment: Ln 129-130: prevalence is expressed as a percentage. You can indicate the number of infected fish in parenthesis.

Authors’ response: the number of infected fish has been indicated in parenthesis

Reviewer comment: How did you define parasite density? This was not explained in the methods.

Authors’ response: the parasite density has been replaced with a better word which is mean intensity. Mean intensity is the average number of parasites in an infected fish.

Reviewer comment: Table 2 does not seem to have a purpose, as you didn’t conduct any tests including salinity and temperature.

Authors’ response: table 2 has been removed. thank you.

Reviewer comment: Ln 147-149: am I understanding correctly that you only observed and measured one out of the 7275 collected parasites? This is definitely insufficient and does not reflect intra-species variation.

Authors’ response: No, each fish was observed and the record was taken. Not one fish was observed. That was an error. The statement has been corrected to reflect Each of the C. clemensi parasites was isolated and viewed on an optical microscope (Figure 1) for the initial minor observations while the detailed observations were observed with Scanning Electron Microscope (SEM) (Figure 2) Thank you.

Reviewer’s response: In this case, you need to provide the number of parasites measured and a table or supplementary data that show some descriptive statistics. As it is, section 3.2. seems to be describing the measurements of one single individual and not a representative number of the collected parasites.

Discussion

Reviewer comment: Ln 203-204: this statement seems redundant, as these are in fact different species.

Authors’ response: the statement “However, these species are quite different from C. clemensi identified in this study” have now been removed since there is no need of stating such again.

Conclusion

Reviewer comment: What is the relevance of your conclusions? From your manuscript alone, it’s not clear to me that the presence of clemensi is harmful to the crimson snapper. How does your study help in the control and management of copepods?

Authors’ response: the conclusion has been rewritten to reflect the identification, prevalence and mean intensity of C. clemensi. The harmful effect of the parasite in terms of the pathology was not part of the objective of this study. However, the harmful effect could be a major justification for this study. Also, once the C. clemensi prevalence and intensity is known, appropriate step to guide and control C. clemensi can be taken.

Author Response

Thank you very much for your efforts in making this paper a wonderful manuscript through appropriate and sincere review. Our responses are highlighted below.

Keywords

Reviewer comment: Avoid repeating words already included in your title.

Authors’ response: some keywords occurring in the title have been replaced, for instance, Aquaculture, fish parasites and Penang, have been included.

Reviewer’s response: “Lutjanus erythropterus” and “Penang” are still present in both title and abstract. I suggest you to avoid repetition so that you can add other keywords that might help you reach more readers through journal databases.

Authors’ response: Lutjanus erythropterus” and “Penang” have now been removed and replaced with “Aquaculture industry” and “Phylogenetic tree” as suggested. Thank you.

Reviewer comment: Please use italics when referring to scientific names.

Authors’ response: all the scientific names have been italicized as suggested, thank you.          

Introduction

Reviewer comment: The introduction does not provide a strong background for the study aim.

Authors’ response: additional information has been provided to buttress the study aim and the additional information is: “Caligus species is responsible for causing massive mortalities in cultured fishes L. erythropterus due to pathological tissue alterations on the gills and buccal cavity thereby leading to serious economic loss. They cause serious harm on the host fishes when they feed on the epidermal tissue, blood and mucus of fishes”. Further justification includes “lack of molecular identification of Caligus clemensi in Malaysia and need for updated information on the spread of the disease. All these have been included to form a strong background for the aim.

Reviewer comment: Use full scientific name when introducing a species for the first time (Genus species) and then use abbreviated form throughout the text (species).

Authors’ response: full scientific names have now been included when introduced for the first time and abbreviated subsequently as suggested. Thank you.

Reviewer comment: 42-44: do you mean that the prevalence was constant throughout the 10 months or was that the study period?

Authors’ response: Thank you for pointing this out. The period of 10 months indicated have been removed. The total prevalence of 81% was an average prevalence. It now read as “Leaw et al. (1) reported a high parasitic infestation rate of 81% on L. erythropetrus fish from floating cages of Penang Island”

Reviewer comment: Ln 49-50: this suggests that this is one of your goals, however you don’t mention anything related to parasite spread later in your manuscript.

Authors’ response: yes, one of our goals is to determine the parasite prevalence. Meanwhile the word “spread” has been replaced by “prevalence”. The prevalence was well highlighted in line 265-282.

Reviewer comment: 51-53: what is the purpose of the molecular identification? You seem to know that you’re dealing with C. clemensi already.

Authors’ response: the purpose of molecular identification in this study is to properly detect the parasites, differentiate and compared to others in the same genus or family. Moreso that molecular diagnosis is very sensitive compared to other methods of diagnosis including microscopy. Also, phylogenetic tree classifying the parasite with respect to others was done.

Reviewer’s response: if so, then perhaps is better that you change “C. clemensi” for “Caligus spp.” in ln 52-53. As it is, it appears you already know the species you’re dealing with and, since you don’t explicitly mention that you want to compare with other Caligus populations, the importance of the molecular identification in this study is not clear.

Authors’ response: “C. clemensi” have replaced Caligus spp.” As indicated. Thank you sir.

Methods

Reviewer comment: When was the study conducted?

Authors’ response: The study was conducted between February to May, 2016 and this is now included in the methodology.

Reviewer comment: Ln 61-62: how were length categories defined? Are there any physiological or ontogenetic reasons behind this?

Authors’ response: Yes, the fish length was categorised based on the ontogenetic reasons. The minimum, average and maximum length were classified accordingly, based on the general size of the fish caught. Thank you.

Reviewer’s response: please clarify which ontogenetic reasons were behind this categorization in the text.

Authors’ response: generally, the fish length was categorised according to physical ontogeny based on growth and development of the fishes with respect to size such that we have the smallest, the medium, and the largest. This has now been clarified in the text. Thank you for your understanding sir.

Reviewer comment: Ln 64-66: where?

Authors’ response: the study was conducted at the Aquatic Research Complex L24, USM, Penang and this has been included in the methodology. Thank you.

Reviewer’s response: my question was regarding the parasite collection. Where were the infested areas of the fish?

Authors’ response: the infested area of the fish were head, body, and both sides of inner operculum, and this has been included in the revised manuscript.

Reviewer comment: Ln 67-68: is it common practice to place sea water fish in fresh water? What is the purpose behind this? Methodology is not sufficiently developed.

Authors’ response: the purpose behind placing sea water fish in fresh water is to anaesthetise the fish since fresh water medium can be used as anaesthetics to reduce the stress as well as for easy handling. This statement has been included in the methodology. Thank you.

Reviewer comment: Ln 69-71: what is the purpose of taking on site temperature and salinity?

Authors’ response: we initially wanted to see the effect of temperature and salinity of the fish but did not have any significant effect. Thus, the statement about the salinity and temperature have been removed. The table has also been removed.

Reviewer comment: 71-75: prevalence is a well understood measure, I don’t think you need to include a formula here.

Authors’ response: formula have been removed. Thank you for your suggestion.

Reviewer comment: Ln 90: what do you mean by “every parasite was separated accordingly”?

Authors’ response: the statement has been removed since it does not have any significant meaning. Thank you

Results

Reviewer comment: Ln 129-130: prevalence is expressed as a percentage. You can indicate the number of infected fish in parenthesis.

Authors’ response: the number of infected fish has been indicated in parenthesis

Reviewer comment: How did you define parasite density? This was not explained in the methods.

Authors’ response: the parasite density has been replaced with a better word which is mean intensity. Mean intensity is the average number of parasites in an infected fish.

Reviewer comment: Table 2 does not seem to have a purpose, as you didn’t conduct any tests including salinity and temperature.

Authors’ response: table 2 has been removed. thank you.

Reviewer comment: Ln 147-149: am I understanding correctly that you only observed and measured one out of the 7275 collected parasites? This is definitely insufficient and does not reflect intra-species variation.

Authors’ response: No, each fish was observed and the record was taken. Not one fish was observed. That was an error. The statement has been corrected to reflect Each of the C. clemensi parasites was isolated and viewed on an optical microscope (Figure 1) for the initial minor observations while the detailed observations were observed with Scanning Electron Microscope (SEM) (Figure 2) Thank you.

Reviewer’s response: In this case, you need to provide the number of parasites measured and a table or supplementary data that show some descriptive statistics. As it is, section 3.2. seems to be describing the measurements of one single individual and not a representative number of the collected parasites.

Authors’ response: 20 C. clemensi were well-observed and measured sir. The measurements have been uploaded as supplementary data. Thank you very much sir.

Discussion

Reviewer comment: Ln 203-204: this statement seems redundant, as these are in fact different species.

Authors’ response: the statement “However, these species are quite different from C. clemensi identified in this study” have now been removed since there is no need of stating such again.

Conclusion

Reviewer comment: What is the relevance of your conclusions? From your manuscript alone, it’s not clear to me that the presence of clemensi is harmful to the crimson snapper. How does your study help in the control and management of copepods?

Authors’ response: the conclusion has been rewritten to reflect the identification, prevalence and mean intensity of C. clemensi. The harmful effect of the parasite in terms of the pathology was not part of the objective of this study. However, the harmful effect could be a major justification for this study. Also, once the C. clemensi prevalence and intensity is known, appropriate step to guide and control C. clemensi can be taken.

Please I like to inform you that the reference numbers have changed due to the additional references suggested to be included in the manuscript. Thank you for your hard work in finetuning this manuscript sir.

Reviewer 2 Report

I think the manuscript has improved and is more readable and clear. However, there are still some points to be corrected /clarified. I detail some of them bellow:  

L16. Remove dot

L21,22. Italics in C. clemensi

L36. italics in Caligus

L36-40. add references to support these statements

L41. Change species for genera

L45. E. fuscoguttatus

L50. Define prevalence and mean intensity. Add references for this statement “Though some findings on the prevalence, mean intensity and veterinary problems 50 caused by Caligus species of copepods has been reported”

L52-60. Suggestion:

Moreover, there is a need for adequate updated epidemiological information on the prevalence of Caligus species in Penang to evaluate the impact of C. clemensi on cultured Crimson Snapper and take appropriate action for its control. With this aim, morphological and molecular identification of C. clemensi on cultured Crimson Snapper were performed to elucidate the parasite prevalence and mean intensity.  In this study, C. clemensi was identified for the first time in Malaysia with species confirmation by molecular techniques.

L64. Use the same font

L65. Add the reference of the legal document about animal ethics in your country

L66. Change “experiment” for “survey”

L69. Fish length

L70. Why this categories? References? Mention the number of fishes in each category.

L72. Reference

L75. Use more recent references in which the procedure you describe is applied.

L76. The total number…

L76-77. “by transferring live fish from seawater to fresh water and performing an exterior scrub on the fish.” What is the difference with the previous procedure? I understand you did not use anesthetic in this time. Why not?

L78. Ref 7: specify the chapter in the reference list; ref 8 use a more recent reference. Also, add the type and brand of the microscope you used.

L87. How many samples?

L80. Remove bold case

L96. How many?

L98, 100. μL

L129. Remove “meanwhile”

L.140. Delete “meanwhile”

L.237. Indicate your p value

L145. Remove (Copepoda:Caligidae)

L146. Did you checked the 7275 specimens isolated? I think you did not mention the optical microscopy observation in the material and method section, please add.

L160. Remove “Leica USA” this should be mentioned in material and methods.

Fig 5. Try to improve the quality of this figure

Table 2. I think this table is not necessary

L196. It is not needed to repeat common and scientific name all the time. Mention them just at the beginning and then you can use the common name throughout the document

L199. Reference

L233. What about other countries?

References: check italics in scientific names

Author Response

Reviewer comment: L16. Remove dot

Authors’ response: dot has been removed

Reviewer comment: L21,22. Italics in C. clemensi

Authors’ response: C. clemensi has been italicised.

Reviewer comment: L36. italics in Caligus

Authors’ response: C. clemensi has been italicised.

Reviewer comment: L36-40. add references to support these statements

Authors’ response: references have been added to support these statements in the revised version. Thank you sir.

Reviewer comment: L41. Change species for genera

Authors’ response: species have now been replaced with genera.

Reviewer comment: L45. E. fuscoguttatus

Authors’ response: now corrected as E. fuscoguttatus

Reviewer comment: L50. Define prevalence and mean intensity. Add references for this statement “Though some findings on the prevalence, mean intensity and veterinary problems 50 caused by Caligus species of copepods has been reported”

Authors’ response: Prevalence and mean intensity are now defined in the manuscript. Prevalence is defined as the percentage total number of infected fishes while mean intensity is defined as the average number of parasites in an infected fish. Also, references have been added. Thank you sir.

Reviewer comment: L52-60. Suggestion: Moreover, there is a need for adequate updated epidemiological information on the prevalence of Caligus species in Penang to evaluate the impact of C. clemensi on cultured Crimson Snapper and take appropriate action for its control. With this aim, morphological and molecular identification of C. clemensi on cultured Crimson Snapper were performed to elucidate the parasite prevalence and mean intensity.  In this study, C. clemensi was identified for the first time in Malaysia with species confirmation by molecular techniques.                                                                                                                                           

Authors’ response: Thank you very much for this wonderful suggestion, it has been included in the manuscript as appropriate.

Reviewer comment: L64. Use the same font

Authors’ response: The same font has been used as suggested. Thank you.

Reviewer comment: L65. Add the reference of the legal document about animal ethics in your country

Authors’ response: the reference has now been added. The reference added is See, A. W. L. (2013). Animal protection laws of Singapore and Malaysia. Sing. J. Legal Stud., 125.

Reviewer comment: L66. Change “experiment” for “survey”

Authors’ response: experiment has been replaced with survey

Reviewer comment: L69. Fish length

Authors’ response: fish length has been included as suggested.

Reviewer comment: L70. Why these categories? References? Mention the number of fishes in each category.

Authors’ response: generally, the fish length was categorised based on growth and development of the fishes with respect to size such that we have the smallest, the medium, and the largest. This has now been clarified in the text. Thank you for your understanding sir., references have been added to the statement under methodology, and the number of fishes in each category have been specified in Table 1. It includes 26-28: 50 (25%), 29- 31: 98 (49%), 32- 35: 52 (26%). Thank you.

Reviewer comment: L72. Reference

Authors’ response: references has been added in the revised manuscript

Reviewer comment: L75. Use more recent references in which the procedure you describe is applied.

Authors’ response: recent references have been included

Reviewer comment: L76. The total number…

Authors’ response: this have been modified. Thank you sir.

Reviewer comment: L76-77. “by transferring live fish from seawater to fresh water and performing an exterior scrub on the fish.” What is the difference with the previous procedure? I understand you did not use anaesthetic in this time. Why not?

Authors’ response: there was no change in the methods. the statement has modified and combined to align. Thank you sir.

Reviewer comment: L78. Ref 7: specify the chapter in the reference list; ref 8 use a more recent reference. Also, add the type and brand of the microscope you used.

Authors’ response: reference 7 and 8 have been replaced with a recent reference, meanwhile the reference number have changed after including additional reference as suggested for some statements. The brand of microscope has been added which is the dissecting microscope (magnification: X50; Leica, USA)

Reviewer comment: L87. How many samples?

Authors’ response: 200 fish samples were examined and have been included in the revised manuscript.

Reviewer comment: L80. Remove bold case

Authors’ response: Bold case have been removed in the revised manuscript

Reviewer comment: L96. How many?

Authors’ response: this have been included, 200 fishes and 7275 parasites. Meanwhile we were only able to critically observed 20 C. clemensi under SEM. Thank you sir.

Reviewer comment: L98, 100. μL

Authors’ response: Now corrected as appropriate. Thank you

Reviewer comment: L129. Remove “meanwhile”

Authors’ response: “meanwhile” have been removed as suggested. Thank you.

Reviewer comment: L.140. Delete “meanwhile”

Authors’ response: “meanwhile” have been removed as suggested. Thank you.

Reviewer comment: L.237. Indicate your p value

Authors’ response: p value has been indicated. Thank you.

Reviewer comment: L145. Remove (Copepoda:Caligidae)

Authors’ response: “(Copepoda:Caligidae)” have been removed.

Reviewer commentL146. Did you checked the 7275 specimens isolated? I think you did not mention the optical microscopy observation in the material and method section, please add.

Authors’ response: No, not all were checked. We checked 20 C. clemensi and took the measurements which have been included in the supplementary data. the optical microscopy observation is now included in the material and method section.

Reviewer comment: L160. Remove “Leica USA” this should be mentioned in material and methods.

Authors’ response: “Leica USA” have now been removed. Thank you.

Reviewer comment: Fig 5. Try to improve the quality of this figure

Authors’ response: we have tried to improve Fig. 5. Thank you very much sir.

Reviewer comment: Table 2. I think this table is not necessary

Authors’ response: Table 2 have been removed as suggested.

Reviewer comment: L196. It is not needed to repeat common and scientific name all the time. Mention them just at the beginning and then you can use the common name throughout the document

Authors’ response: repetition of common and scientific names has been adjusted to reflect only the common names in those parts with repetition.  

Reviewer comment: L199. Reference

Authors’ response: the references have been included, thank you for pointing this out for us.

Reviewer comment: L233. What about other countries?

Authors’ response: other countries including studies conducted in Egypt and Canada have been included and the references added as appropriate.

Reviewer comment: References: check italics in scientific names

Authors’ response: the scientific names in references have been italicised as suggested. Thank you.

Please I like to inform you that the reference numbers have changed due to the additional references suggested to be included in the manuscript. Thank you for your hard work in finetuning this manuscript sir.
